# Association of Mutations Identified in Xanthinuria with the Function and Inhibition Mechanism of Xanthine Oxidoreductase

**DOI:** 10.3390/biomedicines9111723

**Published:** 2021-11-20

**Authors:** Mai Sekine, Ken Okamoto, Kimiyoshi Ichida

**Affiliations:** 1Department of Pathophysiology, Tokyo University of Pharmacy and Life Science, 1432-1 Horinouchi, Hachioji, Tokyo 192-0392, Japan; sekine@toyaku.ac.jp; 2Department of Applied Biological Chemistry, Graduate School of Agricultural and Life Sciences, The University of Tokyo, 1-1-1 Yayoi, Bunkyo-ku, Tokyo 113-8657, Japan; akenokamoto@g.ecc.u-tokyo.ac.jp

**Keywords:** xanthine oxidoreductase (XOR), xanthine dehydrogenase (XDH), xanthine oxidase (XO), xanthinuria

## Abstract

Xanthine oxidoreductase (XOR) is an enzyme that catalyzes the two-step reaction from hypoxanthine to xanthine and from xanthine to uric acid in purine metabolism. XOR generally carries dehydrogenase activity (XDH) but is converted into an oxidase (XO) under various pathophysiologic conditions. The complex structure and enzymatic function of XOR have been well investigated by mutagenesis studies of mammalian XOR and structural analysis of XOR–inhibitor interactions. Three XOR inhibitors are currently used as hyperuricemia and gout therapeutics but are also expected to have potential effects other than uric acid reduction, such as suppressing XO–generating reactive oxygen species. Isolated XOR deficiency, xanthinuria type I, is a good model of the metabolic effects of XOR inhibitors. It is characterized by hypouricemia, markedly decreased uric acid excretion, and increased serum and urinary xanthine concentrations, with no clinically significant symptoms. The pathogenesis and relationship between mutations and XOR activity in xanthinuria are useful for elucidating the biological role of XOR and the details of the XOR reaction process. In this review, we aim to contribute to the basic science and clinical aspects of XOR by linking the mutations in xanthinuria to structural studies, in order to understand the function and reaction mechanism of XOR in vivo.

## 1. Introduction

Xanthine oxidoreductase (XOR) is an enzyme that catalyzes two reactions, the hydroxylation of hypoxanthine to xanthine and xanthine to uric acid, which are the end products of purine metabolism [1,2,3]. XOR takes two forms, xanthine dehydrogenase (XDH, EC: 1.17.1.4) and xanthine oxidase (XO, EC: 1.17.3.2), which reduce NAD^+^ or O_2_ to produce NADH or reactive oxygen species (ROS, H_2_O_2_ and O_2_^−^), respectively. The main function of XOR in vivo is to produce uric acid, and humans, as primates, are prone to hyperuricemia and gout because we evolutionarily lack uric acid oxidase. Therefore, XOR is a target of therapeutic agents for hyperuricemia and gout [4], and XOR inhibitors, allopurinol, febuxostat, and topiroxostat are commonly used in clinical practice. Moreover, in some diseases, the inhibitors are expected to have effects other than uric acid reduction by inhibiting ROS production. The metabolic effects of XOR inhibitors are similar to those of isolated XOR deficiency, xanthinuria type I, which is useful in understanding their effect on metabolism [5,6]. It is a relatively rare inherited autosomal recessive disorder, which is characterized by excessive excretion of xanthine in the urine [7,8,9]. Patients are generally asymptomatic, and are often diagnosed when their serum uric acid levels (less than 1 mg/dL) are measured during physical examinations. The suppression of uric acid production leads to an increase in the uric acid precursor oxypurines (hypoxanthine and xanthine). However, hypoxanthine is salvaged as a substrate for hypoxanthine-guanine phosphoribosyltransferase (HPRT, EC: 2.4.2.8) (Figure 1), so only trace amounts are detected in serum [8,10,11]. On the other hand, urinary excretion of xanthine is markedly increased because guanine is metabolized to xanthine by guanine deaminase (GDA, EC: 3.5.4.3). Xanthine has low solubility and can form xanthine stones, but the patients have no other serious symptoms [8]. Xanthinuria was first identified in 1954 [7], and since then, more than 150 cases have been reported [9]. Since the identification of two mutations in xanthinuria patients by Ichida et al. in 1997 [12], several more mutations have been reported in xanthinuria. Analysis of the pathogenesis of xanthinuria and the enzymatic function conferred by mutations provides useful information for clarifying the role of XOR in vivo. In this review, we outline the XOR reaction mechanism, inhibitors, and XOR deficiency based on the structural and enzymatic findings from mutagenesis and enzyme inhibition studies, and discuss the relationship between mutation sites and enzyme activity.

## 2. Overall Structure of XOR

XOR is a large protein of 300 kDa, consisting of about 1330 amino acids and 2 subunits of about 150 kDa [13,14]. Each subunit consists of three domains: an N-terminal domain containing two [2Fe-2S] clusters (20 kDa), an intermediate domain containing the FAD (40 kDa), and a C-terminal domain containing the molybdenum cofactor (Moco) (85 kDa) (Figure 2). The linker peptide connecting each domain consists of approximately 60 amino acids [14]. XOR exists as XDH in tissues and uses NAD+ as an electron acceptor [15,16], but mammalian enzymes convert it to XO and use O_2_ as an electron acceptor [1,2].

The site that catalyzes the hydroxylation reaction is Moco in the C-terminal 85 kDa domain. Moco has a structure in which the molybdenum atom is bound to the two sulfur atoms of molybdopterin. Molybdenum has several oxidation states, and in oxidized XOR, it exists in the Mo(VI) state. When the enzyme is reduced, Mo(VI) shifts to the Mo(IV) state (Figure 3). In the oxidized form, molybdenum is further coordinated by sulfide (=S), oxo (=O), and hydroxo (-OH), but in the reduced form, Mo=S is replaced by Mo-SH [17,18,19]. The spatial arrangement of Moco and the surrounding amino acid residues in the substrate-binding site is shown in Figure 3A,B. These amino acid residues (Arg881, Glu803, and Glu1262, human sequence) are highly conserved among species and are thought to be involved in the reaction mechanism [20].

In the N-terminal 20 kDa domain, there are two kinds of [2Fe-2S] cluster centers, called Fe/S I and Fe/S II [21,22,23]. Fe/SI is coordinated to the 113Cys-Xaa2-116Cys-//148Cys-Xaa1-150 Cys-motif of the α-helical domain, while Fe/SII is coordinated to the N-terminal-43Cys-X-48Cys-X-51Cys-//73Cys-motif of the ferredoxin-like domain (human sequence) (Figure 2) [21,24]. Fe/SI and Fe/SII are distinguished by their redox potential and EPR signals, with Fe/SII having a higher potential [24]. Electrons passed to molybdenum by hydroxylation are transferred to FAD via Fe/S I and Fe/S II [24,25]. The pterin ring exists between molybdenum and Fe/SI and is involved in electron transfer. Among the redox reaction centers, Fe/SII and FAD are particularly close to each other. Fe/SII is not only involved in the electron transfer pathway to FAD but also functions as an electron sink during enzyme turnover, contributing to the regulation of FAD reactivity [24].

The central 40 kDa domain contains the FAD and NAD binding sites. FAD, which resides in a deep cleft in the domain, exposes the si face of its isoalloxazine ring to the solvent, and NAD can access FAD in the same space [26]. In contrast to the open si face, the re face is in close proximity to the residues of the protein chain, and the side chain of Phe337 (human) is positioned parallel to the isoalloxazine ring (Figure 5A). In its immediate vicinity, there is a highly packed and unique cluster consisting of four amino acids: Arg335, Trp336, Arg427, and Phe550 (human) [27]. They are bound in the XDH form mainly through π-cation interactions. Only Phe550 (human) is located in a long linker between the FAD domain and the Moco domain. In addition, there is an A-loop consisting of Gln423-Lys433 (human) (Figure 5A). In XDH, Asp429 (human) is located closer to the flavin than Arg426 (human) (Figure 5A). These regions around the FAD are involved in the active conversion of the XDH to XO.

## 3. Reaction Mechanism of Uric Acid Production

The elucidation of the hydroxylation mechanism of xanthine to uric acid was aided by the knowledge of the crystal structure of the complex of the XOR inhibitor and the enzyme, which will be discussed later, and is generally considered to proceed as follows (Figure 3). Xanthine enters the active site by forming hydrogen bonds with surrounding amino acid residues so that C8 is oriented toward molybdenum (Figure 3C). The C=O of C2 is in a position to interact with Arg881. Glu803 is protonated and is suitable for forming hydrogen bonds with the NH of N7 and the C=O of C6. Glu1262 takes a proton from Mo-OH, and the protonated Glu1262 forms a hydrogen bond with N9 of xanthine (Figure 3D) [19,28]. The formation of hydrogen bonds between N7 and N9 facilitates the nucleophilic attack on the C8 position of xanthine by Mo-O^−^ (Figure 3D). The H^−^ from C8 moves to the adjacent Mo=S, and Mo=S becomes Mo-SH (Figure 3E) [29]. Two electrons are then transferred to molybdenum [30], reducing Mo(VI) to Mo(IV) [2,31]. The reaction intermediate Mo(IV)-O-C is formed with xanthine, but this lifetime is short, and oxygen molecules are introduced into the substrate by water molecules and replaced by uric acid (Figure 3F). In this process, Mo(IV) is re-oxidized to Mo(VI), and electrons are quickly transferred in the order of Moco, Fe/SI, Fe/SII, and FAD. The electrons transferred to FAD eventually reduce NAD+ or O_2_ to produce NADH, O_2_^−^, or H_2_O_2_.

## 4. Inhibition Mechanisms of the Inhibitors

Allopurinol has a structure in which the C8 and N7 of hypoxanthine are interchanged (Figure 4A). Because it has a purine skeleton, it is hydroxylated to oxypurinol by XOR and aldehyde oxidase (AO, EC: 1.2.3.1), and is also metabolized by other purine metabolizing enzymes, such as HPRT and purine nucleoside phosphorylase (PNP, EC: 2.4.2.1) [32]. Allopurinol functions as a suicide substrate for the enzyme, and the mechanism of XOR inhibition proceeds as follows. The C2 of allopurinol is hydroxylated and converted to oxypurinol. In the course of the reaction, the transiently generated reduced Mo(IV) and N8 of oxypurinol form a covalent bond (Figure 4A) [33,34]. In this process, oxypurinol is thought to form a complex efficiently by rotating in the substrate-binding pocket. Oxypurinol, like xanthine, forms hydrogen bonds with Glu803, Arg881, and Glu1262 (human) (Figure 4A).

Febuxostat is a larger molecule than allopurinol and does not have a purine backbone (Figure 4B). Therefore, its clinical advantage is that it is unlikely to affect other purine-metabolizing enzymes. Febuxostat has a thiazole ring and a benzene ring, and by rotating between these two cyclic structures, it can be better adapted to the structure of the enzyme. As shown in Figure 4B, there are almost no open spaces in the substrate-binding channel. Although febuxostat does not form covalent bonds with molybdenum (Figure 4B), it does form ionic bonds, hydrogen bonds, and π-π interactions with the thiazole ring and nearby phenylalanine residues (Phe915 and Phe1010), as well as van der Waals interactions and hydrophobic interactions [35]. In addition, the CN group present in the febuxostat forms a hydrogen bond with Asn769, which is distant from the active center [36]. Since many weak bonds are formed individually, they become very difficult to separate and show a strong inhibitory effect. The kinetic mode of inhibition of febuxostat is mixed inhibition because the strengths of inhibition of oxidized and reduced XOR are different [35,36].

Topiroxostat has a dipyridine triazole structure and is a suicide substrate inhibitor, in which its C atom of topiroxostat forms a covalent bond with molybdenum via the O atom (Figure 4C) [18,34,37]. This Mo-O-C bond is of the same structure as the reaction intermediate when reacted with xanthine. The structure is stabilized by the exclusion of neighboring water molecules (Figure 4C) [18], which makes the cleavage of the bond less likely to occur [37]. In addition, the CN group, the nitrogen of the unsubstituted pyridine ring, and the nitrogen of the triazole ring are hydrogen bonded to Asn769, Glu1262, and Glu803, respectively (Figure 4C). It also has π-π interactions between the triazole ring and Phe915 and Phe1010 (Figure 4C). In other words, topiroxostat is a hybrid inhibitor that combines the characteristics of allopurinol and febuxostat (Figure 4).

## 5. Potential Effects of XOR Inhibitors

The superiority or inferiority of allopurinol and febuxostat is currently the subject of international interest. According to the results of the Cardiovascular Safety (CARES) study, reported in 2018 [38], febuxostat has a higher risk of all-cause mortality and cardiovascular death, compared with allopurinol, and an alert was issued by the FDA. However, questions have been raised about the methodology of the study, and a number of papers on cardiovascular events with both drugs have since been published. The FAST trial, reported in November 2020 [39], showed that febuxostat is as safe as allopurinol for cardiovascular events, in contrast to the findings of the CARES trial. On the other hand, it has been suggested that inhibition of XOR may have reduced cardiovascular damage and death. In the CARES trial, more than half of the patients stopped taking their medication, and a detailed study including these dropouts showed that the majority of deaths occurred in the absence of medication [40,41,42]. There have been many reports that XOR inhibitors ameliorate cardiovascular disorders, especially based on the association with oxidative stress [43,44,45,46]. XDH in tissues produces ROS by converting to XO under certain conditions, and it has been suggested to have various physiological and pathological roles, such as protection against infections [47] and ischemia-reperfusion injury [1,43,48,49,50]. It has been reported that the XDH form of XOR has NADH oxidase activity, which produces ROS using NADH as a reducing agent [51,52,53]. For example, under conditions of excess NADH, such as during ischemia–reperfusion, XOR can produce ROS using NADH and molecular oxygen as substrates and may be involved in tissue damage. In the cardiovascular system, one of the physiological activities involving XOR is the production of NO, which is a potent vasodilator [54,55,56,57,58]. NO is produced by the reduction of NO_2_ by XO under hypoxic conditions, and it has a tissue-protective effect [59,60,61]. On the other hand, NO traps and scavenges ROS, producing a more potent radical—peroxynitrite (ONOO-) [62]. The decrease in vasorelaxation due to NO inactivation and the cytotoxicity of ONOO- result in endothelial dysfunction [63,64]. Therefore, the use of XOR inhibitors may have a protective effect on tissues during oxidative stress. However, it has been pointed out that too low serum uric acid levels correlate with adverse biological effects, such as increased overall mortality [65,66,67,68]. Lowering uric acid levels, induced by XOR inhibition, may result in oxidative stress injury because uric acid is an antioxidant and acts as a radical scavenger in the body [69,70,71,72,73]. However, there are no reports indicating that the use of currently available XOR inhibitors causes oxidative stress damage. The cases of classical xanthinuria suggest that inhibition of XOR itself does not cause serious adverse effects in humans. The analysis of XOR-deficient patients may be useful in elucidating these detailed mechanisms.

## 6. Conversion of XDH to XO

Reversible conversion of XDH to XO involves the formation of a disulfide bond between Cys536 (human) on the linker connecting the FAD and Moco domains and Cys993 (human) in the Moco domain (Figure 5A,C) [15,74,75,76,77,78,79]. Irreversible conversion also involves limited proteolysis of the same domain linker peptide by proteases, with partial degradation occurring after Lys-552 (human) for trypsin and after Lys-569 for pancreatin [14,77,80]. These trigger the loss of Helix531-535 (rat), which is part of the linker. The less stable linker dissociates from the protein surface. In this process, Phe549 (rat) (Figure 5B), a component of the amino acid cluster on the loop, is displaced, causing the cluster to collapse (Figure 5C) [81]. This opens the solvent gate, which causes the A-loop (Gln422-L432, rat) to move, preventing the approach of NAD+. As the A-loop moves, the amino acid residues around the FAD switch places, and in XO, the negatively charged Asp428 (rat) leaves the isoalloxazine ring and the positively charged guanidinium group of Arg425 (rat) approaches (Figure 5C). This reversal of the electrostatic environment near FAD causes a change in the reduction potential of FAD, which increases its reactivity with oxygen [78,81,82,83,84]. Furthermore, it has been shown that the C-terminal peptide is involved in the conversion of XDH to XO [6]. In XDH, the C-terminal peptide is inserted into the cavity of the FAD site, which has been shown to regulate NAD binding and stabilize the conformation. Kusano et al. [85] generated mice fixed to O and D forms and observed their growth, lifespan, and reproduction, but neither showed obvious phenotypes. After tumor implantation, XO knock-in (ki) mice showed stronger tumor growth than wild-type (WT) or XDH ki mice. It was suggested that XOR, through its XO form, contributes to chondrocyte mineralization and pathological calcification of osteoarthritis cartilage.

## 7. XOR Deficiency

XOR deficiency is classified into three types (Figure 6) [5]. Type I (OMIM #278300) is caused by an isolated deficiency of XOR [12], while Type II (OMIM #603592) is caused by a deficiency of molybdenum cofactor sulfurase (MOCOS, EC: 2.8.1.9) [86]. MOCOS activates XOR enzymatically by adding a sulfur atom [87,88] to one of the Mo ligands of Moco in the complementary molecule family of XOR. AO, an enzyme similar to XOR [89], also requires sulfur atoms as an essential factor, so type II loses the activity of both XOR and AO (two-enzyme deficiency). Type III (OMIM #603707) is caused by an enzyme deficiency in the biosynthetic pathway of molybdopterin and results in Moco deficiency (Figure 6) [90]. In addition to XOR and AO, the activity of sulfite oxidase (SO, EC: 1.8.3.1), containing molybdopterin, is also lost in type III (three-enzyme deficiency). SO does not lose its activity in type II because the sulfur atom of a conserved cysteine residue (Cys207 in human SO) coordinates with Mo and the addition of the sulfur atom by MOCOS is not necessary [91]. The sulfur ligands for XOR and AO are inorganic sulfur, not derived from the protein itself, and require addition in the final maturation step. 

Types I and II are similar in phenotype and cannot be distinguished clinically. These are called classical xanthinuria, and the incidence is estimated to be 1/69,000 [92]. In contrast, the phenotype of type III is completely different and is easily distinguished from classical xanthinuria. SO deficiency in type III causes abnormalities in the oxidation of sulfite, resulting in sulfite accumulation in the mitochondria. In patients, urinary excretion of sulfite is increased, and sulfite accumulates in the brain, resulting in severe neurological symptoms, developmental delay, and lens dislocation [93].

## 8. Detected Genetic Abnormalities

It is known that there are many mutations and single-nucleotide polymorphisms (SNPs) in XOR. The mutations listed in Table 1 have been reported to date, and most of them have been detected in the Moco domain. The amino acid mutation sites are shown on the crystal structure for each domain, according to Table 1 (Figure 7). The human XOR is originally a 1333 amino acid protein, but nonsense mutations that produce early stop codons produce incomplete truncated proteins. Patients with the nonsense mutation in Arg228Ter have normal mRNA levels of XDH, but no enzyme protein [12]. While missense mutations result in a fully folded protein with a normal conformation, mutations in residues that are directly involved in catalysis in the Moco and FAD domains are thought to result in loss of activity. It is also possible to lose activity partially through effects on protein conformation, even if not directly related to catalysis.

Missense mutations in the Fe/S domain affect the formation of the Fe/S cluster structure, and thus electron transfer may not function properly. A case in which the Arg149Cys mutation occurred near a Cys residue in the Fe/S cluster showed type 1 xanthinuria (Figure 7A) [98]. In addition, mutation of the Cys residue, which is a component of the Fe/S cluster, resulted in loss of activity, formation of an unstable monomer, and lack of Moco [24]. Therefore, Cys residues are thought to be important for protein folding, dimer formation, and even Moco insertion [24,113]. Gln102Arg, a missense mutation between Fe/SI and Fe/SII (Figure 7A), is thought to alter the electrostatic environment by substituting arginine with a stronger charge, which may disrupt the structure of the Fe/S cluster and cause misfolding [96].

XOR is also involved in the metabolism of thiopurine drugs such as 6-mercaptopurine (6-MP) [114]. The presence of XOR mutations and the use of inhibitors may increase the blood levels of these drugs, thereby increasing toxicities, such as myelosuppression. Four gene mutations—Thr117Ser, Gly172Arg, Ile703Val, and Arg913Gln—have been reported to be involved in thiopurine intolerance [97]. Kudo et al. [101,106] compared the oxidation of xanthine and 6-thioxanthine (6-TX) using XO mutants. The mutant of Arg149Cys showing type 1 xanthinuria was inactive for both xanthine and 6-TX (Figure 7A) [101,106]. Thr117Ser and Arg913Gln are located near the Fe/S cluster and near the substrate binding site of the Moco domain (Figure 7A,C), respectively, and may be involved in the decrease in activity. The Ile703Val mutant, located on the surface of the Moco domain (Figure 7C), showed an approximately 2-fold higher Vmax than the WT when xanthine was used as a substrate, but a 69.4% lower Vmax when 6-TX was used [101,106]. These contrasting results may be related to the fact that mutations alter the substrate specificity of XOR. His1221Arg has also been reported as a mutation involved in increased activity [101,106], as well as Ile703Val, and is located on the surface of the Moco domain (Figure 7C). Such mutations may increase the rate of uric acid release, since stopped-flow studies with XDH have shown that the rate-limiting step of the whole reaction is uric acid release [2]. Comparison of the crystal structures of mammalian XOR and bacterial XOR revealed a significant difference in the strength of inhibition by the tight binding inhibitor febuxostat, despite the fact that the active center and substrate binding pocket of both enzymes have almost identical structures [115]. The molecular dynamics (MD) simulations showed that the bacterial enzyme was unable to maintain binding to the inhibitor due to the differences in pocket movement between the two enzymes. Residues that affect the reaction rate of the substrate, despite their distance from the substrate binding pocket, may have an effect on the movement of the substrate binding pocket. Gly172Arg is located in the linker peptide that connects the Fe/S domain to the FAD domain and has also been identified as a genetic variant associated with hypertension (Gly172Arg, Ala932Thr, Asn1109Thr) [100]. Ala932Thr and Asn1109Thr are located in the Moco domain (Figure 7C), and both are far from the substrate binding site. The kinetic parameters of Gly172Arg are similar to those of the WT [18]. The linker connecting the Fe/S domain to the FAD domain is conserved among species, but the electron density of the linker is not clear in the crystal structures [14]. If the Gly172Arg mutation is associated with hypertension, XOR may modulate the function of blood vessels and the heart by binding to membrane structures and other proteins through the linker portion. To elucidate the pathogenesis of these missense mutations, enzymatic activity using purified enzymes is essential, and further studies are needed.

## 9. Conclusions

In summary, XOR is an esoteric enzyme with several cofactors that contribute not only to uric acid production but also to ROS generation. The crystal structure analysis of XOR and its inhibitors provided important information on the structure, function, and metabolism of the enzyme. Xanthinuria—an inherited disease that causes hypouricemia—is a good model for studying the effects of XOR inhibitors, and in linking the mutations to structural and functional studies of XOR—some of which are associated with hypertension—it was suggested that the conversion of XDH to XO may regulate vascular and cardiac function. The three XOR inhibitors currently used for the treatment of hyperuricemia and gout are thought to also suppress the production of ROS and NO and may be able to expand the range of indications for the treatment of conditions such as ischemia-reperfusion injury and atherosclerosis in the future. However, the molecular mechanism of tissue protection of XOR inhibitors needs further discussion.

## Figures and Tables

**Figure 1 biomedicines-09-01723-f001:**
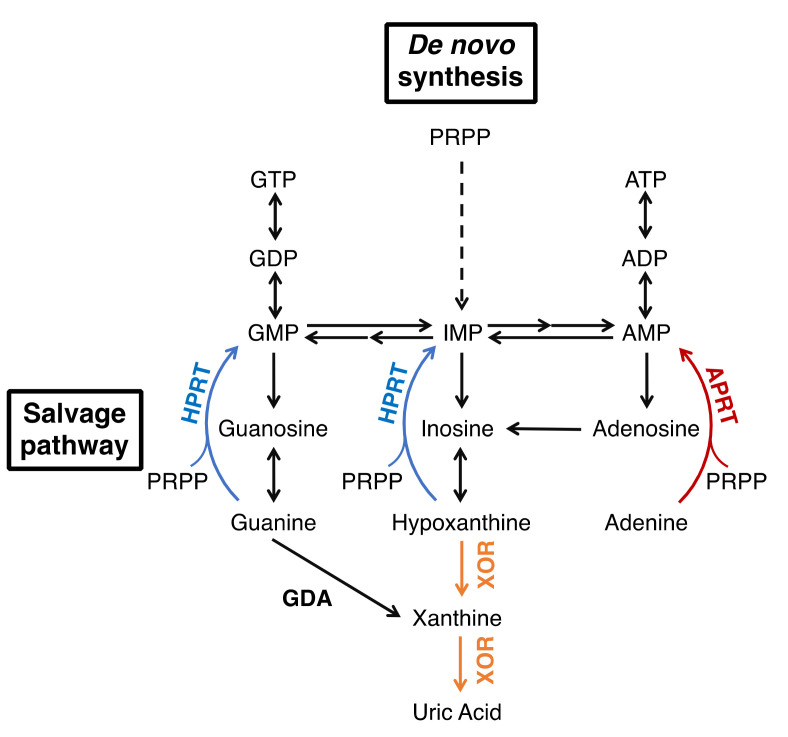
Purine metabolism.

**Figure 2 biomedicines-09-01723-f002:**
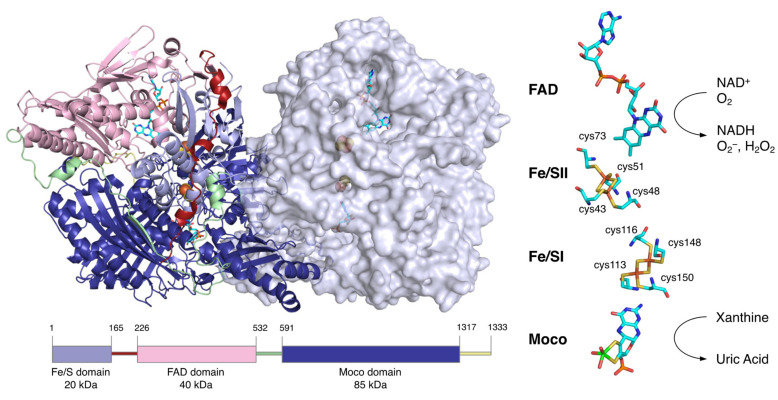
Molecular structure of human XOR. Figures generated from the Protein Data Bank (PDB, ID: 2E1Q). The homodimer structure of human XOR is illustrated with one subunit as a ribbon model and the other as a space-filling model. The N-terminal (light blue), intermediate (pink), and C-terminal (deep blue) domains contain the two [2Fe-2S] clusters, FAD and Moco, respectively. The linker between the Fe/S domain and the FAD domain (red), the linker between the FAD domain and the Moco domain (light green), and the C-terminal peptide (yellow) are shown. A schematic of the domain structure is shown at the bottom in the same color. The cofactor arrangement is shown on the right.

**Figure 3 biomedicines-09-01723-f003:**
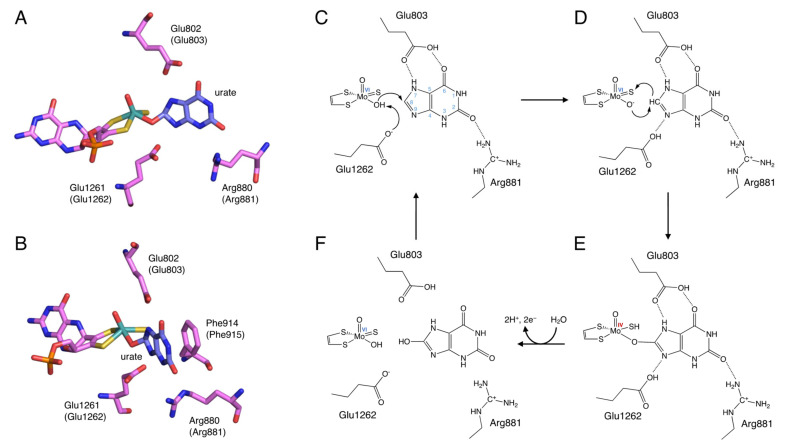
Binding mode of uric acid to XOR and hydroxylation mechanism of xanthine to uric acid. (**A**,**B**) Crystal structures urate bound to XOR (PDB ID: 3AMZ). (**C**–**F**) Scheme of hydroxylation of xanthine to uric acid.

**Figure 4 biomedicines-09-01723-f004:**
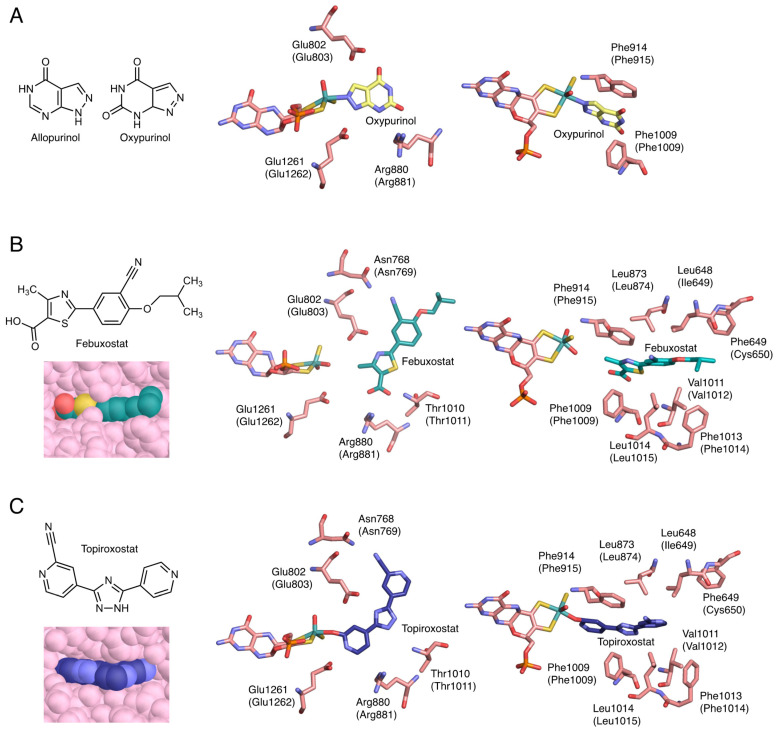
Crystal structure of inhibitors bound to bovine XOR. Human sequences are shown in parentheses. (**A**) Chemical structures of allopurinol and oxypurinol, and the interactions of oxypurinol with surrounding amino acid residues (PDB ID: 3BDJ). (**B**) Chemical structure of febuxostat and the interactions of febuxostat with surrounding amino acid residues and space-filling model (PDB ID: 1N5X). (**C**) Chemical structure of topiroxostat and the interactions of topiroxostat with surrounding amino acid residues and space-filling model (PDB ID: 1V97).

**Figure 5 biomedicines-09-01723-f005:**
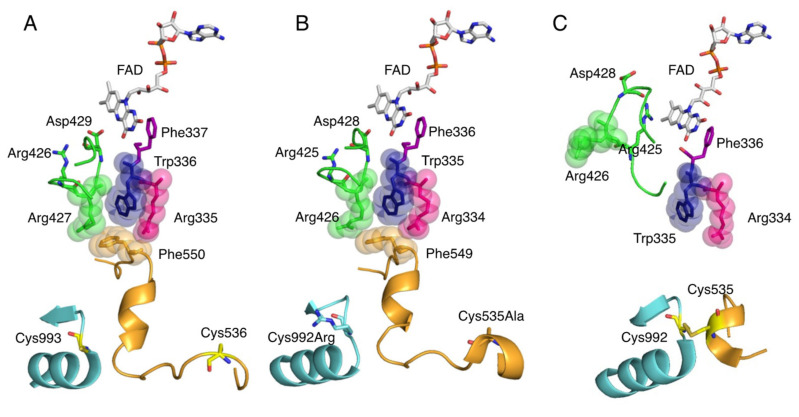
Crystal structure of amino acid clusters involved in conversion of XDH to XO. Active site structure around FAD cofactor. (**A**) Crystal structure of human XOR (PDB ID: 2E1Q), (**B**) rat XDH (PDB ID: 1WYG), and (**C**) rat XO (PDB ID: 4YTZ).

**Figure 6 biomedicines-09-01723-f006:**
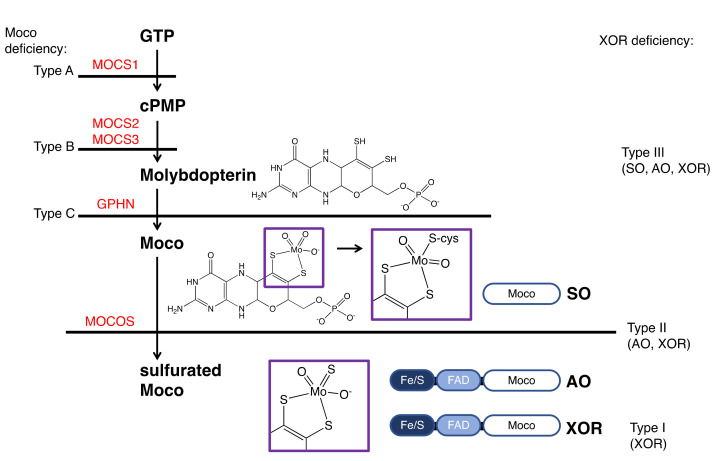
XOR deficiency and the biosynthetic pathway of Moco. The classifications of the types of Moco and XOR deficiency is shown on the left and right, respectively. Enzymes involved in the biosynthesis of Moco are shown in red. Mutations causing Moco deficiency have been identified in the genes MOCS1 (type A), MOCS2 or MOCS3 (type B), and rarely in GPHN (type C). The domain structures of SO, AO, and XOR and the final maturity of Moco are shown. The atoms coordinating to Mo are shown in the purple boxes.

**Figure 7 biomedicines-09-01723-f007:**
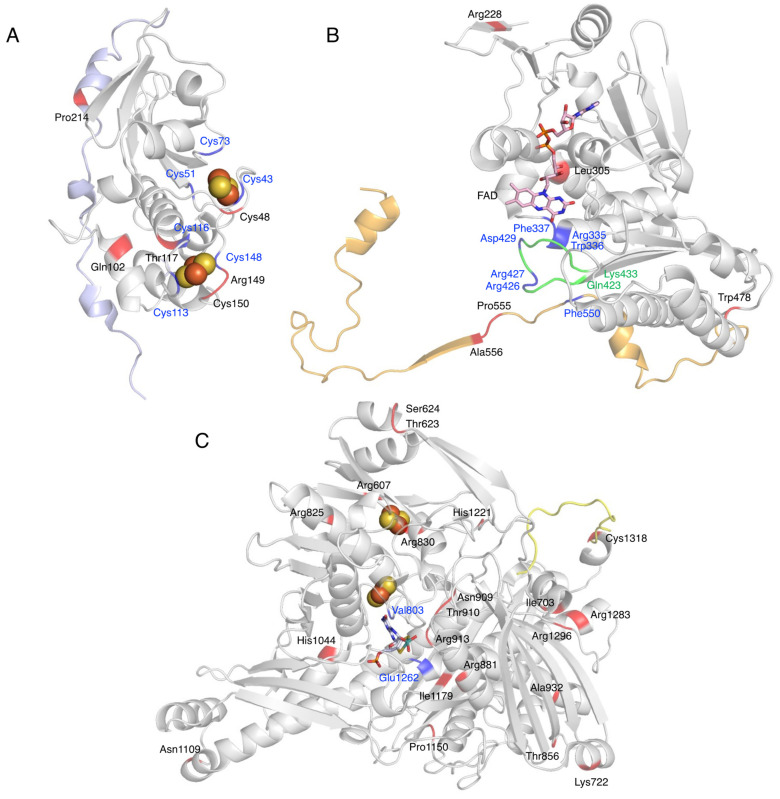
Mutation sites of XOR. Fe/S domain (**A**), FAD domain (**B**), and Moco domain (**C**) of human XOR are shown as a ribbon model. The linker between the Fe/S domain and the FAD domain is shown in light blue (**A**), the linker between the FAD domain and the Moco domain is shown in orange (**B**), and the C-terminal peptide is shown in yellow (**C**). Blue indicates amino acid residues that are important for enzyme function. Mutation sites are shown in red. Green indicates the A-loop consisting of Gln423-Lys433.

**Table 1 biomedicines-09-01723-t001:** Mutation sites of XOR.

Domain	Variant	Phenotype	Reference
Fe/S domain	exon 2–4 del (~11 kbp)	Xanthinuria, type 1	[94]
	p.Cys48Leufs*12	Xanthinuria, type 1	[95]
	p.Gln102Arg	Xanthinuria, type 1	[96]
	p.Thr117Ser	Thiopurine intolerance	[97]
	p.Arg149Cys	Xanthinuria, type 1	[98]
	p.Cys150Phe	Xanthinuria, type 1	[99]
domain linker	p.Gly172Arg	Hypertension	[100]
		Thiopurine intolerance	[101]
	p.Pro214Glnfs*4	Xanthinuria, type 1	[102,103]
	IVS8ds + 1 G > T	Xanthinuria, type 1	[104]
FAD domain	p.Arg228*	Xanthinuria, type 1	[12]
	p.Leu305fs*1	Xanthinuria, type 1	[99]
	p.Trp478*	Xanthinuria, type 1	[99]
domain linker	p.Pro555Ser	Decreased activity	[101]
	p.Ala556Serfs*15	Xanthinuria, type 1	[105]
Moco domain	p.Arg607Gln	Decreased activity	[101]
	p.Thr623Ile	Decreased activity	[101]
	p.Ser624*	Xanthinuria, type 1	[99]
	p.Ile703Val	Increased activity	[101]
		Thiopurine intolerance	[106]
	p.Lys722*	Xanthinuria, type 1	[107]
	p.Arg825*	Xanthinuria, type 1	[103]
	p.Arg830Cys	Xanthinuria, type 1	[108]
	p.Thr856Lysfs*73	Xanthinuria, type 1	[12,109]
	p.Arg881*	Xanthinuria, type 1	[103]
	p.Asn909Lys	Decreased activity	[101]
	p.Thr910Lys	XDH deficiency	[101]
	p.Thr910Met	Xanthinuria, type 1	[95,110]
	p.Arg913Trp	Xanthinuria, type 1	[111]
	p.Arg913Gln	Thiopurine intolerance	[97]
	p.Ala932Thr	Hypertension	[100]
	p.His1044fs*12	Xanthinuria, type 1	[104]
	p.Asn1109Thr	Hypertension	[100]
	p.Pro1150Arg	Decreased activity	[101]
	p.Ile1179Thr	Xanthinuria, type 1	[104]
	p.His1221Arg	Increased activity	[101]
	p.Arg1283Ter	Xanthinuria, type 1	[112]
	p.Arg1296Trp	Decreased activity	[95]
C-terminal	p.Cys1318Tyr	Decreased activity	[101]

## Data Availability

Not applicable.

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
