# Peer review of "Association of Mutations Identified in Xanthinuria with the Function and Inhibition Mechanism of Xanthine Oxidoreductase"

_biomedicines, 2021, doi:10.3390/biomedicines9111723_

Round 1

Reviewer 1 Report

The manuscript entitled “Association of mutations identified in xanthinuria with the function and inhibition mechanism of xanthine oxidoreductase” investigates the structure and activities of XOR through “mutagenesis studies of mammalian XOR and structural analysis of XOR–inhibitor interactions”.

The article is well written and comprehensively describes the details of the XOR molecule and their function, even with the help of explanatory figures. However, it deserves some comments.

Minor points

Line 133

Please, define here the meaning of the abbreviations AO, HPRT and PNP.

Line 184

“XDH in tissues produces ROS by converting to XO under certain conditions, and it has been suggested to have various physiological and pathological roles, such as protection against infections and ischemia-reperfusion injury [1,43,47-50].”

Authors should consider that, under adequate conditions, ROS can also be generated by the NADH oxidase activity of XDH. In addition, please rephrase the period to better distinguish between physiological and pathological roles of XO activity.

Main concern

Conclusions are missing from this review. Authors have to add a chapter with their final considerations “for elucidating the biological role of XOR” as stated in the abstract. The reader would appreciate a comparative discussion about the “three XOR inhibitors currently used as hyperuricemia and gout therapeutics” since they reduce the production not only of uric acid but also of ROS and NO generated by XOR. Finally, the way in which some XOR mutations are associated with hypertension and can modulate the function of blood vessels and the heart deserves further discussion.

Author Response

We would like to thank the reviewers for the insightful comments on our paper. The comments have helped us to improve our paper significantly. Please see the attachment.

Line 133: Please, define here the meaning of the abbreviations AO, HPRT and PNP.

Response: Thank you for pointing this out. The meaning of the abbreviation has been added to the text. 

Line 184: “XDH in tissues produces ROS by converting to XO under certain conditions, and it has been suggested to have various physiological and pathological roles, such as protection against infections and ischemia-reperfusion injury [1,43,47-50].”

Authors should consider that, under adequate conditions, ROS can also be generated by the NADH oxidase activity of XDH. In addition, please rephrase the period to better distinguish between physiological and pathological roles of XO activity.

Response: We have added the information you pointed out from Line 190. 

Main concern

Conclusions are missing from this review. Authors have to add a chapter with their final considerations “for elucidating the biological role of XOR” as stated in the abstract. The reader would appreciate a comparative discussion about the “three XOR inhibitors currently used as hyperuricemia and gout therapeutics” since they reduce the production not only of uric acid but also of ROS and NO generated by XOR. Finally, the way in which some XOR mutations are associated with hypertension and can modulate the function of blood vessels and the heart deserves further discussion.

Response: I added a chapter as a conclusion after Line 402. 

Reviewer 2 Report

The review is well written, and will be important for the full understanding of XOR reaction mechanism, inhibitors, and XOR deficiency based on the structural and enzymatic findings. However, there are minor comments as follows.

(1) Please add future prospect of XOR research such as the relationship between mutation and enzyme activity in the end.

(2) It's easier to understand section 4 (Inhibition Mechanisms of the inhibitors) if the chemical structures of oxypurinol, febuxostat, and topiroxostat are provided.

Author Response

We would like to thank the reviewers for the insightful comments on our paper. The comments have helped us to improve our paper significantly. Please see the attachment.

(1) Please add future prospect of XOR research such as the relationship between mutation and enzyme activity in the end.

Response: Thank you for pointing this out. We have added a chapter as a conclusion after Line 402.

(2) It's easier to understand section 4 (Inhibition Mechanisms of the inhibitors) if the chemical structures of oxypurinol, febuxostat, and topiroxostat are provided.

Response: The chemical structure has been added to Figure 4. The legend and numbers have been revised accordingly.